# Pituitary Apoplexy in a Child with Short Stature and Possible Recent SARS-CoV-2 Infection

**DOI:** 10.3390/diagnostics15121453

**Published:** 2025-06-07

**Authors:** Carmen Gabriela Barbu, Luminita Nicoleta Cima, Marian Andrei, Simona Vasilache, Mihaela Țarnă, Ileana Olguta Rizea, Carmen Sorina Martin, Anca Elena Sîrbu, Simona Fica

**Affiliations:** 1Department of Endocrinology, Diabetes Mellitus, Nutrition and Metabolic Disorders, Carol Davila University of Medicine and Pharmacy, 020021 Bucharest, Romania; carmen.barbu@umfcd.ro (C.G.B.); luminita.cima@umfcd.ro (L.N.C.); carmen.martin@umfcd.ro (C.S.M.); anca.sirbu@umfcd.ro (A.E.S.); simona.fica@umfcd.ro (S.F.); 2Department of Endocrinology, Diabetes Mellitus, Nutrition and Metabolic Disorders, Elias Emergency University Hospital, 011461 Bucharest, Romania; mihaela.tarna@yahoo.com; 3Endocrinology Ambulatory Care Center, Pediatric Hospital, 100336 Ploiesti, Romania; dr.vasilachesimona@yahoo.com; 4Department of Radiology and Medical Imaging, Elias Emergency University Hospital, 011461 Bucharest, Romania; ileanarizea@yahoo.co.uk

**Keywords:** pituitary apoplexy, short stature, COVID-19, aromatase inhibitor therapy

## Abstract

**Background and Clinical Significance:** Pituitary apoplexy is an extremely rare condition in children and adolescents with a rapid onset due to acute hemorrhage, infarction, or both in the pituitary gland. Most frequently, pituitary apoplexy is an asymptomatic or subclinical entity. Few cases of pituitary apoplexy with concurrent SARS-CoV-2 infection or COVID-19 vaccination have been reported. **Case Presentation:** We present the case of a 13-year-8-month-old boy who presented in our pediatric endocrinology department for the evaluation of short stature. He was previously diagnosed with secondary hypothyroidism and was treated with levothyroxine. At admission, clinical examination revealed a height of 141 cm (−2.68 SD/−2.4 SD corrected for mid-parental height), normal weight (60th centile), Tanner-stage G2P1, and delayed bone age. Basal IGF1 was normal, but the tests performed to assess the GH reserve confirmed the GH deficiency (peak GH value 3.11 ng/mL after clonidine/0.95 ng/mL after insulin). The brain MRI revealed a subacute pituitary hemorrhage. Thrombophilia and coagulopathies were excluded by further testing. Anti-SARS-CoV-2 (anti-S-protein IgG) antibodies (>200 BAU/mL) were compatible with COVID-19 infection, indicating a possible association between these two entities. At 3-month follow-up, physical examination showed a 3 cm height gain and advancing pubertal development (G4P2). Newer MRI found changes consistent with resolving hemorrhage. The patient was provided immediately with recombinant human GH and aromatase inhibitor therapy to maximize GH treatment response. During follow-up, the rGH dose was adjusted based on IGF1 values, and after 3 years and 10 months, rGH treatment was stopped, reaching a height of 172.3 cm (−0.51 SD) and surpassing the initial prediction of 164.5 cm. **Conclusions:** Pituitary apoplexy, an even rarer complication in the pediatric population, may be associated with SARS-CoV-2 infection. Further studies are necessary to better understand the intertwining of those conditions.

## 1. Introduction

Pituitary apoplexy is a clinical entity characterized by the sudden onset of hemorrhage, infarction, or hemorrhagic infarction, most commonly of a pituitary adenoma [1]. Depending on the definition used by the authors, the reported incidence for symptomatic pituitary apoplexy varies from 0.6% to 9% and from 14% to 22% for subclinical pituitary apoplexy [2,3].

Various studies and case reports identified numerous risk factors for this condition, such as the following: systemic hypertension; major surgery; coagulopathies; dynamic pituitary function tests with TRH, GnRH, CRH, or insulin; estrogen therapy; pregnancy and post-partum states; various medications; radiation therapy; head trauma; pituitary surgery; and gamma knife therapy [4,5,6,7,8,9,10,11,12,13].

In children and adolescents, due to the overall rarity of pituitary adenoma in this age group, accurate information regarding pituitary apoplexy is lacking, and the reported incidence varies from 1.8 to 7.5% of all surgically treated pituitary adenomas. Given the unclear definition of pituitary apoplexy, there is no clear consensus about its characteristics in this age group [3,14].

Recently, during the COVID-19 outbreak, pituitary apoplexy cases in association with this infection have been reported, mostly in the adult population [15]. Almost every case of this kind has an underlying etiology, such as a pituitary adenoma. However, there are case reports with no pre-existing risk factors [16,17].

## 2. Case Presentation

We report the case of a 13-year-8 month-old boy who presented in our endocrinology department due to short stature in November 2021. He was previously evaluated by his territorial endocrinologist and had a repeated slightly lowered FT4 with normal TSH levels and multiple low IGF1 determinations. This prompted his physician to prescribe 25 μg of levothyroxine a day and refer him to our clinic for further investigations. The child was delivered by C-section at 37 weeks, had a birth weight of 3.4 kg, a birth length of 50 cm, and a normal psychomotor development. At admission, his height was 141 cm (−2.68 SD or −2.4 SD if corrected for mid-parental height SDS), weight was 39 kg (60th centile), and head circumference was 59 cm (98th centile), and he had an upper segment to lower segment ratio of 1.09. His pubertal development was G2P1 with a bone age of 12 years. Biological evaluations showed no abnormality in routine blood tests and liver function, normal adrenal and gonadal function, and a prolactin level of 9 ng/mL. Thyroid function was normal with an FT4 level of 1.14 ng/dL and a TSH of 1.69 μU/mL on 25 μg of levothyroxine. Basal IGF1 was detected to be within the normal range, with a value of 116.1 ng/mL (normal values for ages 64–508), and after a provocative test, the peak GH value was 3.11 ng/mL after the oral administration of clonidine and 0.95 ng/mL after an insulin administration of 0.1 UI/kg iv. Adrenal response during ITT was partially present, with a peak cortisol level of 18.1 μg/dL. Before prescribing treatment with recombinant human GH, we performed a brain MRI, which revealed subacute pituitary hemorrhage, as seen in Figure 1. The posterior pituitary bright spot was present, and the pituitary stalk was a normal size.

Ophthalmological testing showed no visual deficits or oculomotor alterations, and having no clinical symptoms of pituitary apoplexy, transsphenoidal surgery was not necessary. This prompted us to discharge the patient with only 25 μg levothyroxine/day and 3-month follow-up MRI. Screening tests for coagulopathies were recommended, and the results were negative. Focusing on the etiology of the pituitary apoplexy, we initially considered Rathke cleft cysts or sellar/suprasellar tumors. Subsequent MRI examination found changes consistent with resolving hemorrhage, weakening the hypothesis for a Rathke cleft cyst. However, physical examination revealed an almost 3 cm height gain with an advanced pubertal development (G4P2). According to this new data and the Bone Expert final height prediction of 164.5 cm, the patient was provided immediately with recombinant human GH 0.03 mg/kg/day, and to maximize GH treatment response, we utilized 2.5 mg of letrozole, based on published case reports. For normal thyroid function, the levothyroxine dose was adjusted to 50 μg/day, and during follow-up, it was adjusted to 100 μg/day. Also, we measured anti-SARS-CoV-2 (anti-S-protein IgG) antibodies (>200 BAU/mL) after the mother mentioned that the boy had a respiratory infection before being first admitted to our pediatric endocrinology department for testing the GH reserve that might have been responsible for the subacute pituitary hemorrhage. During the 3-year follow-up, he was reexamined every 6 months, adjusting the rGH dose based on IGF1 values and body weight, as summarized in Table 1.

Note the IGF1 fluctuation during the follow-up period. It is well known that in healthy children, during pubertal development, IGF1 rises significantly, reaching levels even in the acromegalic range [18]. This is explained by the fluctuating levels of gonadotropins (FSH and LH) and testosterone, transient insulin resistance, and the concurrent rise in IGF-binding protein 3 (IGFBP-3) and the acid-labile subunit (ALS) [19]. Even though we have a GH-deficient patient, these physiological hormonal variations should explain the IGF1 fluctuation. We asked the laboratory department to retest the November 2023 blood sample, and the result was similar, excluding a testing error.

After approximately 2 years of treatment, letrozole was stopped, and in November 2024, rGH treatment was also stopped, given the bone age of 16 years. The patient’s height during the 3 years and 10 months of rGH treatment is shown in Figure 2, reaching 172.3 cm (−0.51 SD) at the last evaluation and surpassing his initial height prediction of 164.5 cm.

As seen in Figure 3, the latest MRI investigation performed in March 2025, as well as other MRI investigations during follow-up, revealed the presence of hemoglobin breakdown products located within the anterior lobe of the pituitary gland, strengthening the hypothesis of a pituitary apoplexy without an adenoma.

## 3. Discussion

Pituitary adenomas are generally the main cause of pituitary apoplexy. Pituitary tumors may produce ophthalmologic symptoms such as visual field defects most frequently, a sudden loss of vision, ophthalmoplegia, or papilledema in rare cases [5,21]. In the case of an acute pituitary hemorrhage or apoplexy, the most common and prominent symptom is headaches, often accompanied by nausea and vomiting. They manifest due to dural traction or the extravasation of blood and necrotic material into the subarachnoid space [3,22,23]. Our patient did not complain of any visual impairments or headaches. However, he presented with signs of pituitary insufficiency, like delayed growth and secondary hypothyroidism. The most life-threatening and frequently encountered hormonal complication is corticotropic deficiency, evolving with severe hypotension and hyponatremia [3,22,24]. In our case, this complication was absent.

The hypothalamic–pituitary gonadal axis functioned properly in our patient, leading to the normal progression of puberty with advancing bone age. Therefore, the most pressing matter, from an endocrinologist’s point of view, was the GH-deficient short stature with an advanced pubertal development, therefore predicting a rapid epiphyseal fusion and a shorter adult height. Having this in mind, we started the rGH treatment, and to delay growth plate fusion, the options were, according to literature data, to associate either a GnRH analogue or an aromatase inhibitor [25].

Suppressing normally timed puberty with a GnRH analogue presents a series of disadvantages, like creating a hypogonadal state at a critical time during development with diminishing whole-body protein synthesis and increasing adiposity. Furthermore, the child will not only be short but also sexually immature compared to his friends, leading to psychological implications. On the other hand, aromatase inhibitors block estrogen and increase testosterone production, leading to a slower growth plate fusion rate and increasing growth and lean body mass in males [25]. Having this in mind, together with the results of several studies and case reports, we decided to associate aromatase inhibitor therapy to maximize rGH treatment response by blocking estrogen production and delaying epiphyseal fusion [26,27,28]. As a result, after almost 3 years of rGH and 2 years of aromatase inhibitor treatment, our patient’s height improved, surpassing the initial predicted height by almost 10 cm and reaching a value close to the 50th percentile.

According to data we found in the literature, in the majority of cases, pituitary apoplexy happens in an underlying pituitary tumor [29]. In our case, the first MRI we have is after the apoplexy developed, and no pituitary adenoma was visible at that point in time; therefore, initially, we could not be certain if our patient had a preexisting adenoma. During follow-up, MRI examination was repeated every 6 months, with the last one in March 2025, with all of them describing no pituitary tumor, confirming the absence of a pituitary adenoma.

Another situation related to pituitary apoplexy in pediatric age is acute hemorrhage of a Rathke cleft cyst (RCC) [30]. Even though RCC hemorrhage can mimic pituitary apoplexy and baseline MRI cannot differentiate between the two entities, a subsequent MRI examination after 3 or 4 months should reveal a stable or growing lesion in the case of RCC hemorrhage, whereas in the case of a pituitary adenoma, it reveals tumor shrinkage [31]. In our case, the first MRI reexamination, carried out after 3 months, found changes consistent with resolving hemorrhage, standing against the diagnosis of RCC hemorrhage. Furthermore, no MRI examination during follow-up revealed a sellar/suprasellar tumor. Pituitary biopsy would have been the technique suited to settle the diagnosis dilemma; however, given the patient’s young age, the absence of any clinical symptoms, and surgical indications, pituitary biopsy was not carried out [32].

Lymphocytic hypophysitis, another condition that can mimic pituitary apoplexy, was excluded based on MRI examination. According to Kurokava et al., the most frequent findings of lymphocytic hypophysitis are loss of the posterior pituitary T1-weighted bright spot and enlarged pituitary stalk. Our patient had none of these findings, standing against this diagnosis [33].

Many other risk factors are described in the literature [29,34]. Some studies indicate clotting disorders as a precipitating factor. Anticoagulated states, whether from taking anticoagulant drugs, thrombolytic agents, or thrombocytopenia and even von Willebrand disease or clotting disorders due to liver disease, can lead to pituitary apoplexy, as emphasized by numerous reports [4,7,35]. Our workup excluded coagulopathies, and we obtained negative thrombophilia screening tests.

Some authors have demonstrated that stimulatory tests, like TRH, CRH, or GnRH testing, might be implicated in generating an apoplexy [8,9,11]. Even the insulin tolerance test is listed as a risk factor in some reports [36]. The exact mechanism is unknown. However, it was speculated that these tests may produce a vasoactive effect that initially leads to ischemia and infarction and later evolves to hemorrhage [7]. Other risk factors listed in the literature, like systemic hypertension, major surgery, estrogen therapy, pregnancy and post-partum states, various medications (dopamine receptor agonist, isosorbide, chlorpromazine, GnRH agonist, and clomiphene), radiation therapy, head trauma, pituitary surgery, and gamma knife therapy, can be easily excluded in this case [29]. Although this is very uncommon, having no previous imaging, we cannot rule out the possibility of a stimulating-test-induced pituitary apoplexy.

Recently, the novel coronavirus disease and SARS-CoV-2 vaccination have been recognized as risk factors for an apoplectic event [34,37,38,39,40]. Although most of the cases described are on a preexisting pituitary adenoma, there are reports of pituitary apoplexy attributed to COVID-19 in the absence of an underlying lesion [17,39,41,42].

While its most frequent clinical manifestation is respiratory distress, which varies from an asymptomatic carrier state to severe respiratory distress syndrome, there are many reports describing extrapulmonary involvement, with pituitary apoplexy being one of them [43,44,45].

The mechanism of such pathology might be hematological and immunological changes induced during infection [46]. The SARS-CoV-2 virus is now known to access neurons and glial cells using the ACE2 receptor and can cause a variety of clinical manifestations such as headache, ataxia, stroke, epilepsy, neuropathic pain, or myopathy. Those neurological symptoms have been attributed to immune-mediated mechanisms and are more likely in patients with severe cases of disease [47,48]. As it was demonstrated that the pituitary cells present ACE2 receptors, the invasion and direct cell death might be a plausible mechanism [49,50]. Aliberti et al. demonstrated, using immunohistochemical analysis in a case of PA, that SARS-CoV-2 proteins were present next to pituitary blood vessels, indicating that antigen cross-reactivity may play a role in inducing PA alongside the immune-mediated glandular damage secondary to antibody formation or cell-mediated damage [51,52]. Regarding the association between COVID-19 and coagulation abnormalities, data from the literature suggest that while the von Willebrand factor (vWF) is upregulated, ADAMTS13, a metalloprotease that cleaves high-molecular-weight vWF, is downregulated. This unbalanced ratio leads to the hypercoagulable state encountered in severe COVID-19, contributing to thrombosis in pituitary gland blood vessels and causing pituitary apoplexy [53,54].

Other hematological changes caused by COVID-19 are characterized by elevated fibrinogen and D-dimer levels, mild prolongation of PT/aPTT, and mild thrombocytopenia. Although COVID-19 is known more as a hypercoagulable state than a bleeding one, spontaneous hemorrhage may complicate a disease-induced thrombosis. Similarly to how pituitary apoplexy is associated with pregnancy, some authors propose an association between a hypercoagulable state and the apoplectic event; we can extrapolate to a link between the hypercoagulable state seen in COVID-19 patients and the occurrence of pituitary apoplexy [48,54,55,56].

Our patient exhibited a symptomatic respiratory infection 4 to 8 weeks before the first admission. Unfortunately, we have no confirmation of a SARS-CoV-2 infection with PCR or rapid antigen test, as it was a mild form of upper respiratory tract infection regarding respiratory symptoms that required no medical consultation. Taking into account this period of 1–2 months, we considered that the shift in antibody titers had happened, and no IgM measurements were carried out. However, we have a very high anti-SARS-CoV-2 (anti-S-protein IgG) antibody level. Numerous studies conducted during the pandemic have assessed the diagnostic accuracy of COVID-19 serological tests, and according to a 2023 meta-analysis, the reported median specificity was over 97% for many antibody tests, resulting in high specificity, particularly when IgG was tested [57]. Although we acknowledge the limitations of serological testing, particularly the potential for antibody cross-reactivity, in the absence of a positive PCR result and given the time elapsed since the suspected infection, serology was our only available method to support prior SARS-CoV-2 exposure. Overall, given the high antibody level and the strong specificity reported for IgG-based serology, there is a high probability that the upper respiratory tract infection in question was indeed COVID-19. Taking this into account, and in the absence of any other identifiable cause for pituitary apoplexy, this case strongly suggests a possible association between COVID-19 and the observed pituitary disease.

There is no consensus on whether early surgery should be performed. An appropriate treatment should be selected from several options, such as surgery (emergency, urgent, or elective) and supportive medical therapy, for each patient based on severity and timing [58,59,60].

Many studies have recommended that neurosurgical decompression should be the first-line treatment, especially in cases with serious neurologic deficits [61]. Others stated that there is no difference between the outcomes of surgical or conservatively treated patients and promoted a conservative management in patients with mild or no symptoms [62,63,64]. Our patient did not present any of these symptoms associated with pituitary apoplexy, and the diagnosis was settled only after imaging studies, so we opted for conservative management and hormonal replacement therapy.

Even though a number of cases of pituitary apoplexy have been reported as a manifestation of SARS-CoV-2 in the adult population, to our knowledge, this is the first pediatric case of pituitary apoplexy associated with SARS-CoV-2 [65]. Ohata et al. conducted a literature search in 2021 regarding cases of pituitary apoplexy in the pediatric population since 1980. They found 65 cases of pediatric pituitary apoplexy, all of them secondary to an adenoma, with most of them comprising macroadenoma [66]. Since 2021, other pediatric cases have been published, the majority of which had a pituitary adenoma, with one apoplexy secondary to RCC hemorrhage and one case with Langerhans cell histiocytosis who developed apoplexy after surgery [67,68,69,70]. Comparing our case with other pediatric cases of pituitary apoplexy, the main difference is the severity of the clinical symptoms. While most of the cases present with visual field defects and headache most frequently or a sudden loss of vision and ophthalmoplegia, often accompanied by nausea and vomiting and requiring decompression surgery, our patient had none of these, and he was brought in for a check-up after hormonal deficiencies manifested themselves. Regarding hormonal deficiencies, corticotropic deficiency was the most frequent hormonal complication in published apoplexy cases; however, in our patient, this complication was not present. What makes this case unique is the absence of a pituitary adenoma, with the description of changes consistent with resolving hemorrhage at every MRI investigation since diagnosis and the remaining link with a possible recent SARS-CoV-2 infection at that moment, suggesting an association between COVID-19 and the pituitary apoplexy.

## 4. Conclusions

Pituitary apoplexy is a rare entity, especially in the pediatric population. It may occur in patients with a preexisting pituitary adenoma or with several risk factors, including coagulopathies or dynamic pituitary function tests. Recently, pituitary apoplexy cases in association with the SARS-CoV-2 infection or COVID-19 vaccination have been reported, mostly in adults. We report the first known pediatric case of pituitary apoplexy in which a potential temporal association with COVID-19 was observed; however, causality could not be confirmed histologically. Even though we have no confirmation of pituitary damage through biopsy, which is possibly related to a SARS-CoV-2 infection confirmed by serology, this case contributes to the growing body of evidence suggesting a possible association between COVID-19 and pituitary hemorrhage.

## Figures and Tables

**Figure 1 diagnostics-15-01453-f001:**
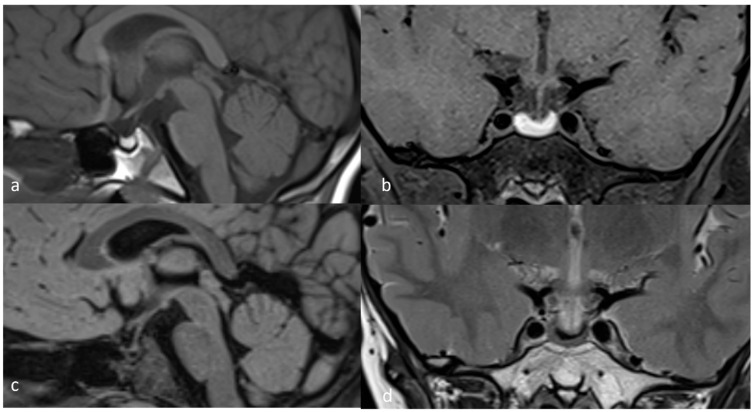
November 2021 MRI scan: (**a**) T1 TSE (turbo spin echo) sagittal; (**b**) T1 TSE FS (fat-suppressed) coronal; (**c**) FLAIR sagittal; (**d**) T2 TSE coronal.

**Figure 2 diagnostics-15-01453-f002:**
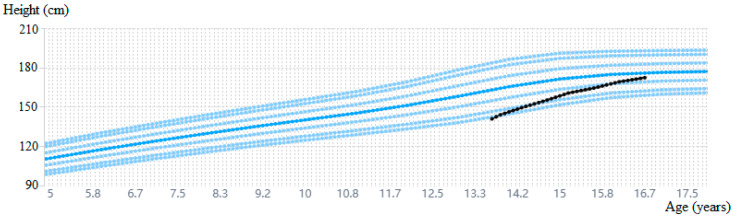
Growth history during follow-up with a measurement error range of 0.5 cm [20].

**Figure 3 diagnostics-15-01453-f003:**
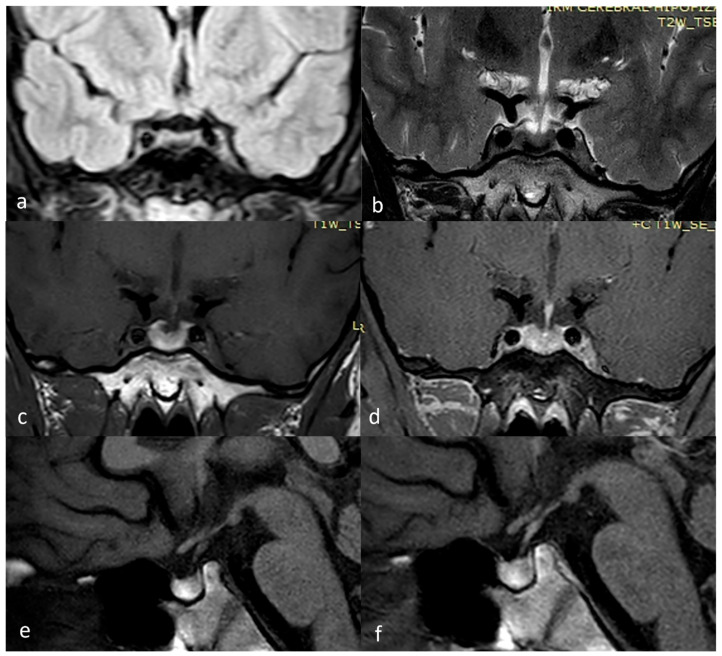
March 2025 MRI scan: (**a**) FLAIR coronal; (**b**) T2 TSE coronal; (**c**) T1 TSE coronal; (**d**) T1 TSE FS coronal +Gd; (**e**) T1 TSE sagittal; (**f**) T1 TSE sagittal +Gd.

**Table 1 diagnostics-15-01453-t001:** Treatment-related characteristics during follow-up.

	IGF1 (ng/mL)/ (Z-Score for Tanner)	Height (cm)/(SD)	Weight (kg)	rGH Dose (mg/kg/day)	Letrozole (mg/day)
November 2021	116.1/−2.18	141/−2.68	39	-	-
January 2022	141.7/−3.25	144.1/−2.42	40.8	Start 0.03	Start 2.5
February 2022	257.4/−1.8			0.03, ↑ to 0.04	2.5
March 2022	504.9/+0.98	145.7/−2.38	42.2	0.04	2.5
June 2022	387.9/+0.16	149.5/−2.10	44.5	0.04, ↑ to 0.045	2.5
December 2022	337.6/−0.45	155.3/−1.82	51	0.045	2.5
May 2023	298.4/−0.96	160.6/−1.46	57	0.045, ↑ to 0.05	2.5
November 2023	633.3/+3	164.4/−1.26	61	0.05, ↓ to 0.045	2.5
May 2024	482.9/+1.44	169.2/−0.83	69.3	0.045	2.5, STOP
November 2024	525.1/+2.26	172.3/−0.51	69.8	0.045, STOP Bone age 16 years old	

↑—dose increase; ↓—dose decrease.

## Data Availability

Data supporting the published results are available from the corresponding author upon request.

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
