# Peer review of "Pituitary Apoplexy in a Child with Short Stature and Possible Recent SARS-CoV-2 Infection"

_diagnostics, 2025, doi:10.3390/diagnostics15121453_

Round 1

Reviewer 1 Report

Comments and Suggestions for Authors

This article provides valuable case data on pituitary apoplexy related to COVID-19 in children, but it needs to include more methodological details and mechanism discussions to enhance persuasiveness.

The study is the first to report a case of pituitary apoplexy following COVID-19 infection in children, partially filling a gap in the literature. However, it does not clearly identify the direct pathological mechanisms between COVID-19 and pituitary apoplexy, such as viral invasion, immune-mediated factors, or coagulation abnormalities. It only hypothesizes based on temporal association and requires further experimental validation.

Single-case studies cannot rule out coincidental factors; thus, similar cases or animal models need to be included for verification. Additionally, the study does not compare with other pediatric pituitary apoplexy cases, making it difficult to distinguish the specific impact of COVID-19.

The positive COVID-19 antibody result only indicates a history of infection without detecting viral RNA or dynamic changes in antibody titers, failing to prove a direct causal relationship.

The cited references are mainly from before 2021, lacking recent key findings from the past three years, which need to be supplemented.

The results section only describes the data, and the discussion does not deeply analyze the potential mechanisms between COVID-19 and pituitary apoplexy, such as coagulation abnormalities or ACE2 receptor expression.

In Figure 2, the growth curve lacks error range annotations, and the IGF1 values in Table 1 fluctuate significantly (e.g., 298.4 ng/ml in May 2023 vs. 525.1 ng/ml in November 2024), requiring an explanation for physiological plausibility.

The IGF1 value of 633.3 ng/ml in November 2023 (Z-score +3) far exceeds the mean, necessitating exclusion of testing errors or the influence of puberty hormone fluctuations.

Reviewer 2 Report

Comments and Suggestions for Authors

The case is stated to be possibly the first pediatric case of pituitary apoplexy associated with SARS-CoV-2. Please clarify whether similar pediatric cases exist in the literature and strengthen the discussion with comparative analysis.  Consider emphasizing why this case is unique. Is it the timing post-infection, the absence of adenoma, the management strategy, or the growth outcome?

The authors mention a possible pre-existing adenoma but also exclude this based on imaging. The distinction between pituitary apoplexy secondary to adenoma versus primary hemorrhage of a normal gland is essential. Clarify the differential diagnosis more firmly. Consider including pituitary apoplexy mimics like RCC or lymphocytic hypophysitis more explicitly in the differential diagnosis.

The case hinges on the connection with SARS-CoV-2, yet there is no PCR or antigen test to confirm the infection. The high IgG level is indirect. How the authors can conclude this case is related to SARS-CoV-2 infection?

Ensure recent pediatric literature is included, if available, especially on growth modulation and COVID-19–related endocrine changes in children.

Round 2

Reviewer 1 Report

Comments and Suggestions for Authors

I have carefully reviewed the revisions submitted by the authors and essentially responded positively. They are now in good standing. However, it is worth noting that some aspects still need improvement and enhancement. For instance, the authors recognise the limitations of PCR deletion and propose a 'strong association' by combining a high antibody titer (>200 BAU/ml) with the exclusion of alternative causes. However, the term 'strong causation' should be avoided (the original 'strongly suggesting an association' is acceptable), and the conclusion should make it clear that this is a 'speculative association'.

Author Response

Response to Reviewer 1 Comments

1. Summary

Dear Reviewer,

Thank you so much for the constructive appraisal comments.

We are pleased to resubmit our revised manuscript entitled “Pituitary apoplexy in a child with short stature and recent SARS-COV-2 infection”

Please find the detailed comments below and the corresponding revisions highlighted in red

in the manuscript:

2. Questions for General Evaluation

Reviewer’s Evaluation

Response and Revisions

Does the introduction provide sufficient background and include all relevant references?

Yes

Is the research design appropriate?

Yes

Are the methods adequately described?

Yes

Are the results clearly presented?

Yes

Are the conclusions supported by the results?

Can be improved

3. Point-by-point response to Comments and Suggestions for Authors

Comments 1: However, it is worth noting that some aspects still need improvement and enhancement. For instance, the authors recognise the limitations of PCR deletion and propose a 'strong association' by combining a high antibody titer (>200 BAU/ml) with the exclusion of alternative causes. However, the term 'strong causation' should be avoided (the original 'strongly suggesting an association' is acceptable), and the conclusion should make it clear that this is a 'speculative association'.

Response 1: Thank you for your constructive concerns. We updated the manuscript avoiding the term “strong causation”. We are glad that you agree with us about this phrase “strongly suggesting an association”.

Furthermore, we addressed the issue of the specificity of serology tests.

We updated the conclusion section according to your recommendation.

Reviewer 2 Report

Comments and Suggestions for Authors

The authors have not adequately addressed my primary concern regarding the proposed relationship between SARS-CoV-2 infection and pituitary apoplexy.
Moreover, their conclusion that SARS-CoV-2 infection was the causative factor in this case is not sufficiently supported, particularly as it relies solely on IgG serology. This method lacks specificity and carries a risk of cross-reactivity with antibodies from other pathogens, making it an unreliable basis for establishing causality.

Author Response

Response to Reviewer 2 Comments

1. Summary

Dear Reviewer,

Thank you so much for the constructive appraisal comments.

We are pleased to resubmit our revised manuscript entitled “Pituitary apoplexy in a child with short stature and recent SARS-COV-2 infection”

Please find the detailed comments below and the corresponding revisions highlighted in red

in the manuscript:

2. Questions for General Evaluation

Reviewer’s Evaluation

Response and Revisions

Does the introduction provide sufficient background and include all relevant references?

Yes

Is the research design appropriate?

Not applicable

Are the methods adequately described?

Not applicable

Are the results clearly presented?

Must be improved

Are the conclusions supported by the results?

Not applicable

3. Point-by-point response to Comments and Suggestions for Authors

Comments 1: The authors have not adequately addressed my primary concern regarding the proposed relationship between SARS-CoV-2 infection and pituitary apoplexy.
Moreover, their conclusion that SARS-CoV-2 infection was the causative factor in this case is not sufficiently supported, particularly as it relies solely on IgG serology. This method lacks specificity and carries a risk of cross-reactivity with antibodies from other pathogens, making it an unreliable basis for establishing causality.

Response 1: Thank you for your constructive concerns. Indeed, the SARS-CoV-2 infection confirmation is based on the high level of IgG antibodies. Please, keep in mind that the first visit to our clinic was 4-8 weeks after the upper respiratory tract infection episode so this was our only way to prove that it was COVID. Having only mild symptoms, the mother did not take him to a doctor to test with RT-PCR, so no test was done at that moment (acute phase). Given that almost 2 months have passed since we first saw the patient, our only choice to prove a possible association with COVID was IgG serology.

We understand you might be concern about the specificity of this method and the risk of cross-reactivity. However, many studies were done during the pandemic years about diagnostic accuracy of COVID-19 tests and we want to mention this 2023 meta-analysis (Vilca-Alosilla, J.J.; Candia-Puma, M.A.; Coronel-Monje, K.; Goyzueta-Mamani, L.D.; Galdino, A.S.; Machado-de-Ávila, R.A.; Giunchetti, R.C.; Ferraz Coelho, E.A.; Chávez-Fumagalli, M.A. A Systematic Review and Meta-Analysis Comparing the Diagnostic Accuracy Tests of COVID-19. Diagnostics 202313, 1549. https://doi.org/10.3390/diagnostics13091549) that clearly concluded “It was discovered that serological tests had a very poor sensitivity but a high specificity, particularly when IgG was found.” The reported median specificity was over 97% in every IgG serological test available in this meta-analysis.

We understand that the PCR test offers greater accuracy, and we wish we had a positive PCR result at the time of infection to provide a strong confirmation.

We updated this statement to clarify any concern about the specificity risk of cross-reactivity: “However, we have a very high anti SARS CoV-2 (anti S-protein IgG) antibodies level. Numerous studies conducted during the pandemic have assessed the diagnostic accuracy of COVID-19 serological tests and, according to a 2023 meta-analysis, the reported median specificity was over 97% for many antibody tests, resulting a high specificity, particularly when IgG was tested[57]. Overall, given the high antibodies level and the strong specificity reported for IgG-based serology there is a high probability that the upper respiratory tract infection in question was indeed COVID-19. Taking this into account, and in the absence of any other identifiable cause for pituitary apoplexy, this case strongly suggests a possible association between COVID-19 and the observed pituitary disease.”

Additionally, we revised the conclusion section to clarify that the suggested association is based on the exclusion of alternative causes and the close proximity with SARS-CoV-2 infection.

Round 3

Reviewer 2 Report

Comments and Suggestions for Authors

Dear Authors,

Thank you very much for your efforts and for your response.

However, in your report titled "Pituitary apoplexy in a child with short stature and recent SARS-CoV-2 infection", the phrasing suggests that the child was definitively diagnosed with a recent SARS-CoV-2 infection. It is important to note that IgG antibodies are not considered reliable markers for diagnosing recent SARS-CoV-2 infection, as they may persist long after exposure and can also result from cross-reactivity.

Please refer to the literature on cross-reactive IgG responses to SARS-CoV-2 in individuals previously exposed to other pathogens, including common viruses, bacteria, or even certain vaccinations. These findings underscore the limitation of using IgG alone as evidence of recent infection.

Therefore, we recommend revising your conclusion. Instead of stating "recent SARS-CoV-2 infection", a more accurate phrasing would be "possible recent SARS-CoV-2 exposure" or "pituitary apoplexy in a child with serologic evidence of prior SARS-CoV-2 exposure", to avoid overinterpretation.

You should revise the conclusion as well.

Round 4

Reviewer 2 Report

Comments and Suggestions for Authors

Thank you for your revision